# A qualitative analysis of the nurturing care environment of families participating in Brazil's *Criança Feliz* early childhood program

Laura Mendes Toledo Dal'Ava dos Santos[1], Lidia Godoi[2], Beatriz de Andrade e Guimarães[3], Isabela Mendes Coutinho[3], Nathalia Pizato[4], Vivian S. S. Gonçalves[5], Gabriela Buccini[6]*

1 Postgraduate Program in Human Nutrition, University of Brasilia, Brasília, Federal District, Brazil, 2 Postgraduate Program in Public Health, University of São Paulo, São Paulo, Brazil, 3 Graduate in Nutrition, University of Brasilia, Brasilia, Federal District, Brazil, 4 Department of Human Nutrition, University of Brasilia, Brasilia, Federal District, Brazil, 5 Graduate Program in Public Health, University of Brasilia, Brasilia, Federal District, Brazil, 6 Department of Social and Behavioral Health, School of Public Health, University of Nevada, Las Vegas, Las Vegas, NV, United States of America

* gabriela.buccini@unlv.edu

**Data Availability Statement:** A de-identified data set is not possible to provide due to ethical and legal considerations. These sharing restrictions are

## Abstract

Investing in early childhood programs such as Brazil's *Criança Feliz Program (PCF)* to support low-income families in providing a nurturing care environment is critical to ensure that children reach their full developmental potential. We aimed to analyze the influence of the PCF on the nurturing care environment provided by families enrolled in the program in the Federal District, Brazil. A qualitative case study was conducted based on in-depth interviews with a purposive sampling of 22 caregivers enrolled in the PCF for at least six months. Eighteen subthemes emerged from the thematic analysis following the five components of the Nurturing Care Framework (i.e., *good health*, *adequate nutrition*, *responsive care*, *early learning*, *and security and safety*). Caregivers recognized the benefits of the PCF on children's mental health (*good health*) and reported challenges in providing adequate nutrition due to food insecurity (*adequate nutrition*). A bond between the home visitor and families was identified as critical to promote responsive parenting practices (*responsive care*). Caregivers appreciated the early stimulation activities provided during PCF home visits (*opportunities for early learning*). Access to social welfare programs, such as conditional cash transfer and food assistance, were facilitated through PCF multisectoral actions (*safety and security*). On the other hand, families reported not receiving support from PCF for issues such as breastfeeding, maternal mental health, and disciplinary practices. In summary, PCF enhanced the components of the nurturing care environment provided by families. However, their vulnerabilities and contextual implementation barriers may prevent families from fully benefiting from PCF activities.

## Introduction

Providing nurturing care for early childhood development (ECD) has become a global priority to break the cycle of poverty and promote social justice [1, 2]. Optimal ECD is shaped by the

imposed by the University of Nevada, Las Vegas Institutional Board Review (IRB). The authors declare that a de-identified data set from this study are available upon request directly to UNLV IRB (irb@unlv.edu).

**Funding:** L. M. T. D. Santos received scholarship from the National Council for Scientific and Technological Development (CNPQ). VSSG received support for the data analysis of the interviews carried out in the Federal District from the Federal District Research Support Foundation (FAPDF) under number 498/2021 - FAPDF/SUCTI/ COOTEC. GB received support for designing and conducting this study from the Eunice Kennedy Shriver National Institute of Child Health & Human Development of the National Institutes of Health under Award Number R00HD097301. The funders had no role in study design, data collection and analysis, decision to publish, or preparation of the manuscript. The content is the sole responsibility of the authors and does not necessarily represent the official opinion of the funders.

**Competing interests:** The authors have declared that no competing interests exist.

quality of the nurturing care environment provided during the early years of a child's life. Nurturing care refers to a stable environment that ensures children's good health and nutrition, protection from threats, and provides opportunities for learning through emotionally supportive and responsive relationships [1, 2]. Efforts cutting across the five components of the Nurture Care Framework (NCF) (i.e., good health, adequate nutrition, responsive care, opportunities for early learning, and safety and security) can help children to thrive and prevent adverse childhood experiences such as physical and emotional abuse, neglect, and household dysfunction [2, 3].

Fostering strong and responsive parenting relationships between children and their caregivers is critical for promoting a nurturing care environment. Responsive parenting is defined as the ability of caregivers to recognize and support the mental, emotional, and physical needs of their children [4]. The use of responsive parenting practices (i.e., less use of coercive and negative communication) enables children's socioemotional, cognitive, and motor development; increases affection; and creates closer bonds and trust between children and their caregivers [5, 6]. Responsive interactions between caregivers and the child through play and early stimulation activities have been proven to lead to better ECD outcomes [7]. Therefore, investing in early childhood programs, such as Brazil's *Criança Feliz Program (Programa Criança Feliz*, PCF*)*, to foster responsive practices can support families in providing a nurturing care environment and ensure children reach their full developmental potential.

PCF was created in 2016 in response to the Legal Framework for Early Childhood in Brazil–the largest country in South America with about 215 million inhabitants. PCF aims to provide home visits and complementary multisectoral actions to promote positive parenting practices and facilitate the access of families to social welfare programs [8]. The home visit adopts the Care for Child Development methodology targeting pregnant persons, children up to 3 years of age in a situation of social vulnerability, and children up to 6 years of age with disability enrolled in the Continuous Cash Transfer Program (CCTP) [8]. The PCF has been scaled up to 3,028 Brazilian municipalities (corresponding to 54% of all municipalities), becoming one of the largest home visiting programs globally [9]. Contextual barriers during the PCF implementation have been documented [10] and have potentially influenced the lack of quantitative impact of the program in improving ECD [11]. However, the qualitative benefits of the PCF on the nurturing care environment provided by families have not been documented.

Among the participating municipalities, the Federal District (Distrito Federal, DF) is one of the largest urban centers with high poverty levels and race/ethnicity inequities located in the Central-West region of Brazil. The PCF implementation in the DF began in 2019 and currently follows up 3,200 low-income families. Implementation barriers in the DF were found to be similar to other municipalities with large urban inequities [12]. Therefore, amid these barriers, the DF can provide an important case study to understand the qualitative benefits of the PCF in large urban centers in Brazil. Thus, we aimed to analyze the influence of the PCF on the nurturing care environment provided by families enrolled in the program in the DF, Brazil.

## Methods

### Study design and ethical aspects

This qualitative case study consisted of in-depth interviews with families enrolled in the PCF in the DF, Brazil. To report this study, we followed the Consolidated Criteria for Reporting Qualitative studies (COREQ) [13].

The study received ethical approval from the Ethics Committee of the University of Brasilia, Faculty of Health Sciences (CAAE: 32390620.0.0000.0030) and the Human Subjects

Institutional Review Board of the University of Nevada Las Vegas. Following a description of the aim and design of the study, all participants provided verbal informed consent, which was audio recorded at the beginning of the interview. The verbal consent was approved by both the Brazilian Ethics Committee and UNLV IRB. All participants received a virtual copy of the consent form.

## Study setting

The DF encompasses the municipality of Brasilia, which is the capital city of Brazil. Located in the Central-Western region of the country, the DF has a large population of 3,010,881 inhabitants residing across 35 administrative regions. In 2021, 57.3% of the population identified as Black or Brown race/skin color, and 15.7% of the population lived in poverty (with an income below US$ 5.50 *per capita* per day) [14, 15]. The PCF was implemented in two phases. The first phase in 2019 included eight administrative regions, and the second implementation phase began in August 2021, adding eight more regions, for a total of 16 regions implementing the PCF. Currently, PCF has 3,200 families enrolled in the program across the 16 regions (1,600 in 8 regions in phase 1 and 1,600 in 8 more regions in phase 2). Details about PCF implementation are described following the Template for Intervention Description and Replication (TIDieR) [16] in S1 Appendix.

## Selection of the participants

We interviewed pregnant persons and caregivers of children under 36 months old. The inclusion criteria were (i) to be enrolled in the PCF for at least 6 months and (ii) to live in one of the eight administrative regions of phase 2 implementation (Brazlândia, Fercal, Gama, Itapoã, Paranoá, Planaltina, Sobradinho, and Varjão). A purposive sampling approach was employed. Eligible families were identified through a list of persons enrolled in the PCF. The research team selected families randomly from the list and contacted the supervisor and home visitor assigned to each selected family, who then granted authorization from the family to share their information contact with the research team. A trained research assistant contacted the families by phone to explain the research purpose, interview procedures, and share a virtual copy of the consent form. If the participant agrees to be interviewed, the research assistant scheduled a convenient time to follow up the interview. No participants refused to participate in the study and the IDIs were conducted virtually in May 2022. The sample size was based on information saturation during preliminary coding and analysis [17].

## Interview guide development

The in-depth interview (IDI) guide was developed in Portuguese by three coauthors (LDS, LG, and GB) based on the five components of the NCF: good health (related to the health and well-being of children and their caregivers), adequate nutrition (related to maternal and child nutrition), responsive care (related to the caregiver's ability to perceive, understand, and respond to their child's signals in a timely and appropriate manner), opportunities for early learning (related to any opportunity for the baby or child to interact with a person, place, or object in their environment), and security and safety (related to safe and protected environments for children and their families). Questions encouraged participants to share their experiences and perceptions about the nurturing care environment provided to their children whether or not linked to the PCF activities (S2 Appendix). When necessary, additional probes were used. Paraphrasing was used to verify that the interviewer had understood and adequately interpreted the responses of the participants. The interview guide also included sociodemographic questions and validated two-item questions to identify risk for household food insecurity [18].

## Data collection and management

Twenty-two IDIs were conducted in Portuguese by a coauthor (LG), who is a native Portuguese speaker, had prior experience with IDI techniques, and had no prior relationship with the participants. The IDIs were conducted virtually between May 2 and May 6, 2022, and lasted between 20 and 50 minutes each. The IDIs were audio recorded after permission was granted, and a native Portuguese-speaking professional transcribed the interviews verbatim. Then, the first author (LDS) compared all transcriptions to the original audio recording to ensure accuracy.

## Data analysis

Data were analyzed in Portuguese by four members of the research team (LDS, BG, IC, GB), who had no prior relationship with the participants. The research team was trained in both qualitative analysis and the components of the NCF.

The transcriptions were codified using a thematic analysis to identify the perceptions of families enrolled in the PCF in relation to the nurturing care environment [19]. The coding process started with the four team members independently reading a transcription and deductively coding it and reaching consensus on the appropriate themes and subthemes within the components of the NCF. This process was repeated with additional five transcriptions when thematic saturation was reached, as the coding of the final two interviews yielded no additional themes [16]. The coding process resulted in a codebook with themes and subthemes mapped across the NCF, and it was used by the first author to code the remaining 12 transcripts. The codebook was iteratively refined and vetted by the research team multiple times to ensure transparency and agreement on code organization. This process generated a final codebook that was used by a single coder to code all interviews using a consistent coding process [20]. Fig 1 outlines the data analysis steps to iteratively refine the codebook. Finally, the quotes that most accurately represent the themes were identified, and these were translated into English by a professional translator. To maintain the confidentiality of the participants, all names have been omitted from the direct quotes; (HV) is used to replace the name of the home visitor, (CH) replaces the name of the child, and (CG) indicates that the name of the caregiver interviewed.

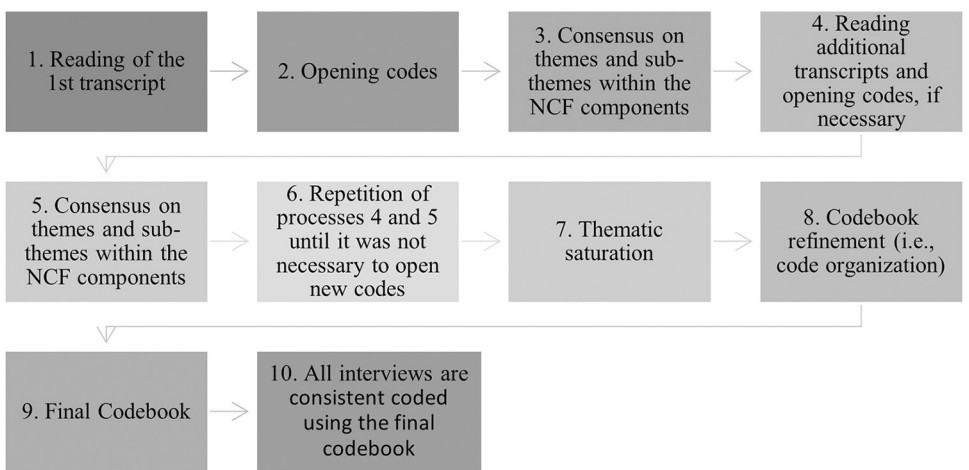

**Fig 1. Illustration of data analysis steps to create and refine the codebook.**

## Results

Twenty-two caregivers were interviewed, including 21 mothers and 1 father. Among them, 15 reported having Brown skin color (68%), 17 completed high school or more (77%), and 18 were unemployed or away from work without pay (81.8%). Most of the families interviewed (71%) had children between 12 months and 35 months of age enrolled in the PCF, and 29% were pregnant women between 24 and 46 years old. Families received on average two social welfare benefits granted by the federal or local government. About 82% of the families were at risk of food insecurity.

Eighteen subthemes influencing the nurturing care environment provided by families were identified across the five components of the NCF. Subthemes are summarized in Fig 2, where the first level refers to the activities of the PCF, and the second level is the external context (i.e., community and other public services not related to the PCF).

### Good health

The subthemes identified within good health related to the benefits of the PCF on the child's mental health, a missed opportunity to promote multisectoral actions with the health sector,

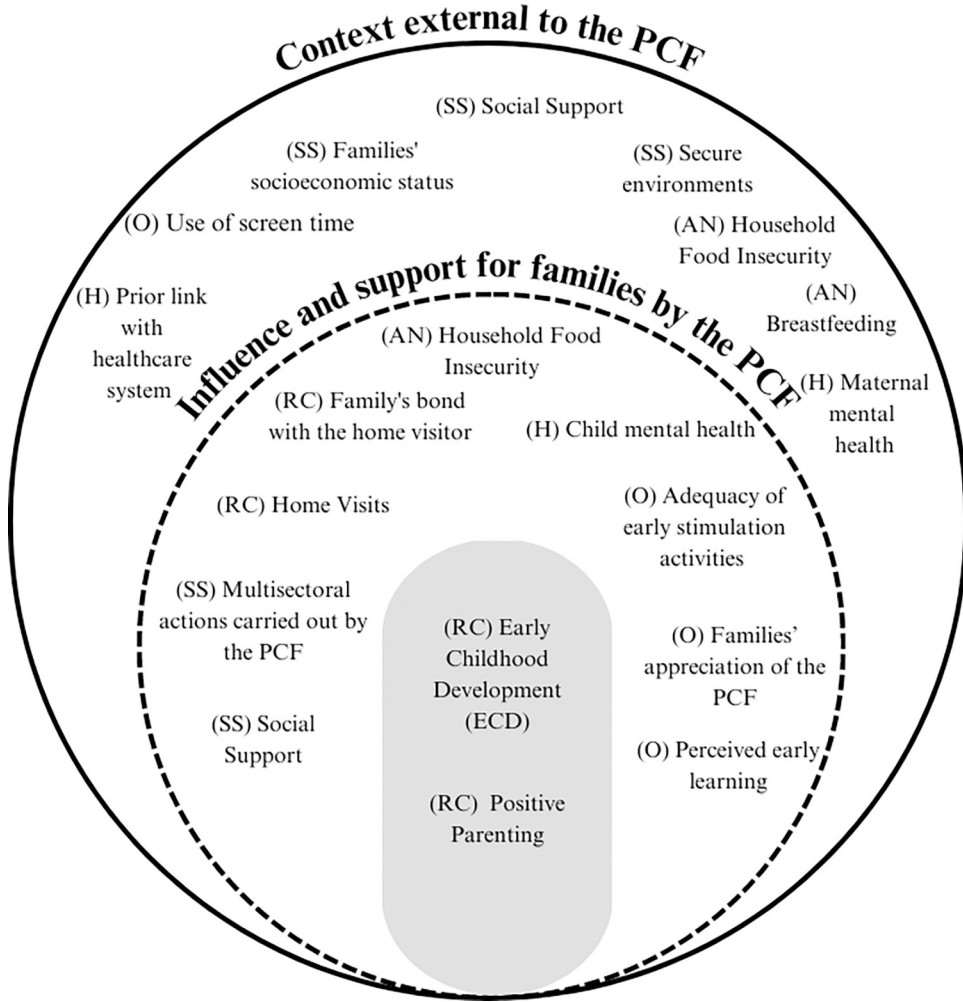

**Fig 2. The influence of the external context and Brazil's *Criança Feliz* program on the nurturing care environment.** (H)–Good Health; (AN)–Adequate Nutrition; (RC) Responsive Care; (O)–Opportunities for Early Learning; (SS)–Safety and Security.

and the complexity of maternal mental health. One of the health benefits of participating in the PCF activities mentioned by caregivers was the promotion of the child's mental health. In general, caregivers perceived home visits as a time in which the child distresses through play, generating appreciation for the presence of the home visitor, as reported by the following caregiver:

*"Wow, it's very nice [the PCF program], when the home visitor comes, especially during the [COVID-19] pandemic. My son likes the activities a lot, and [he] also distresses a little bit." (DF_24).*

All the caregivers reported accessing primary healthcare services prior to enrolling in the PCF. The most cited health services were attending well-child visits to monitor growth and vaccination, as well as prenatal care visits, when pregnant. Children with some disability reported participating in specialized therapies offered by tertiary hospitals. However, there were no reports of joint actions between the PCF and the health care teams, even in the face of complex demands such as the mental health of caregivers. The following quote exemplifies an opportunity for joint intervention between the PCF and health teams:

*"Because my pregnancy was high-risk, I started a treatment with a psychiatrist. I have depression and anxiety, but I have been getting better because I'm taking the medicine for about two weeks. I never asked for [help]. You know, like, when you are tired already? I know that everyone has problems, that everyone has their own life. Even with people, sometimes they ask: '(CG), is everything okay?' Even though I am not okay. . . I sometimes say, 'I'm fine.' Not to bother, you know?" (DF_44)*

In fact, many caregivers reported that motherhood negatively affected their own mental health and quality of life. More than half of interviewees reported exhaustion, loneliness, social exclusion, and anxiety due to the responsibilities of their children and the household chores that they assumed alone. At least two caregivers reported that part of this exhaustion was due to sleep deprivation and the difficulty of their children to sleep through the night, as illustrated below:

*"Very [overwhelmed and stressed]. Mostly because my daughter doesn't sleep every night. She wakes up five times a night [. . .] I say 'my God, she'll be two years old, and she has not changed yet'[. . .] My priority is my daughter; and the house [chores], I only do if I can or I'll fix it later [. . .]." (DF_38)*

Issues related to child development, such as sleep (or sleep deprivation of caregivers), were themes that the interviewees did not feel they could seek support and guidance during the PCF visits or from the health teams.

## Adequate nutrition

The subthemes reported by the families were difficulties in providing adequate nutrition amid food insecurity, the need for complementary multisectoral actions of the PCF, and a missed opportunity to support families on exclusive breastfeeding. Food-insecure caregivers reported that they worry about not having enough food due to financial constraints. In this context, one of the perceived benefits of participating in the PCF reported was having the home visitor as a source of help to obtain food baskets either through donations or referral to a community organization such as churches or community centers or referral to a social welfare program.

The following quote exemplifies the situation of families and the support they received from home visitors:

> "There was a really bad time here at our home. There wasn't anything [to eat]. It was right on the PCF visit day [. . .] But I said: 'It's my son, I'm going to ask for help for my son,' then I shared the situation with the PCF home visitor. He said: 'Don't worry. I'll be right back.' It was true. The PCF home visitor really came back and brought a food basket with him." (DF_24)

Another issue widely mentioned by caregivers was regarding child nutrition, specifically breastfeeding or the lack thereof. Early weaning was frequently reported and attributed to a variety of factors, including lack of support and information regarding breastfeeding. The early introduction of the bottle was often accompanied by the early use of child formula, a new pregnancy, and the indication to interrupt breastfeeding, the mother's employment, and the fact that the mother felt trapped. Half of the caregivers reported not receiving support and guidance about the importance of exclusive breastfeeding or on how to overcome breastfeeding difficulties during the PCF visits. The following statement exemplifies one of the difficulties encountered by a caregiver in relation to breastfeeding and the lack of support:

> "I breastfed my son for only until three months. [. . .] Because he was born premature and he didn't suck well. He was hungry and cried a lot at night, and I didn't know what was happening. One time, he was crying a lot, and I thought it was colic, but I noticed that he didn't have a full stomach. Then, I had regular milk in the house [. . .], and I put it in a little bottle and I gave it to him. He drank it all, slept through the night. I said, 'Oh, well, I'm going to start giving it to him.' Then, I bought the formula that was the right milk for him" (DF19)

## Responsive care

The subthemes reported by the families were the importance of the family connection with the home visitor, execution of the home visits, child development, promotion of positive parenting, and a missed opportunity to promote positive disciplinary practices.

Caregivers mentioned that the home visitors were flexible and sensitive in understanding the situations experienced by the family. In this context, caregivers reported establishing a trusting relationship with home visitors. The trusting relationship was attributed to the feelings that the home visitors care about the children as well as care and welcome caregivers through conversations and questions about their well-being, helping them in challenging times, including times of depression and fear without shaming them. The following quote exemplifies the relationship established between the home visitor and the family:

> "(HV) is wonderful [. . .]. When I'm sad, she comes. She talks to me and cheers me up. Then, we do the activities with my baby and he likes it. Every Thursday, when (HV) arrives at the gate, my baby has a big smile for her [. . .] In addition, during the days of the week without a visit scheduled, she calls me and asks if everything is ok, asks if (CH1) is also ok, if (CH2) is ok." (DF_27)

According to eight caregivers, home visitors adjust the scheduling and the delivery mode of home visits according to the availability of the family (e.g., delivering virtual home visits when the child is attending daycare). At least two families reported that the fathers were absent during the home visits. Families recognized the dedication of the home visitors in the preparation

of the activities and the materials used during the home visit. The consistency of weekly visits at pre-established times and days as well as the consistency of having the same home visitor over time were reported to help increase the trust and bond. Several families expressed the desire for the child to continue participating in the PCF even after reaching the maximum age. About five caregivers wished to increase the frequency of the home visits and expressed their preference for in-person home visits compared to virtual/remote. In addition, caregivers appreciated the flexibility of the dynamics of the home visit to, for example, let the child perform the activity when they feel more comfortable. The following quote exemplifies the recognition of the work performed by the home visitors:

*"I really appreciate the visits from (HV). Sometimes, daily, you don't bother to do certain games, activities [with your child]. Then, (HV) brings these activities to you, and you realize 'wow, that is true, this will lead to my child development.' So, every visit I am very grateful for the (HV) and for her dedication. (HV) is creative [. . .] I see that (HV) makes everything from cardboard material to plastic bottle and brings everything to our home. My child is enchanted by very simple things, but it captures his attention." (DF_38)*

The education activities on positive parenting practices delivery through lectures and meetings at the Social Assistance Reference Center (*Centro de Referência de Assistência Social*, CRAS) by the PCF staff were considered critical to develop the understanding that the way parents deal with their children can influence their future. Thus, caregivers reported understating their critical role in promoting their child ECD. In this context, caregivers reported that the PCF taught them to use positive discipline practices such as praising, encouraging, talking things through, and not using harsh punishment, for example:

*"I don't say (to my daughter) that she can't go, that she can't do it. I say, 'be careful, go, you are a winner, you can do it.' My daughter said, 'Mom, I can't do it,' and I said 'Stop this, you can do it. You can do it. You're brave.' [. . .] Then, she does and succeeds. [. . .] That is what the mother at CRAS said. If I don't tell your daughter that she can, that she's able to, she won't get it into her head. [. . .] I don't have to teach her, make her afraid, or hit her for her to learn something. The child learns not because she's beaten. She learns when you talk to her." (DF_53)*

On the other hand, a few caregivers did not feel prepared to deal with challenging behaviors, such as tantrums, and reported using practices, including shouting and swearing. Four caregivers reported using screens such as television and phones as resources to calm their children. The following quote reveals the options used by caregivers to calm and entertain their children to free the caregiver to perform other domestic activities:

*"We end up saying to the child 'ah, stay here quiet a little bit, take the cell phone,' and we end up giving the cell phone. 'Turn on the TV,' unfortunately. I didn't want to have given her screens when she was two, but, on a day-to-day basis, that didn't happen" (DF_38)*

## Opportunities for early learning

The subthemes reported by the families were the adequacy of the activities proposed by the home visitors, the promotion of the child's learning, and the appreciation of the families of the PCF.

Caregivers acknowledged home visitors' efforts in adapting activities according to the interest of each child to increase opportunities to learn. They appreciated when home visitors follow up after the home visit to remind them to engage with the activities proposed during

home visits. In fact, most caregivers reported carrying out the activities proposed by the home visitor at different times beyond the day of the visit. When replicating the activities, caregivers reported sending text messages to communicate with home visitors about the performance of their child, including strengths or difficulties encountered. However, three families reported barriers that prevent the children from repeating the activities at home, such as the lack of materials at home, the large number of children in the household. Similarly, when parents work outside of the home and a third caregiver participates in the PCF home visit activities, parents did not feel prepared to repeat the activities. These examples illustrate the need for extra guidance and support to adapt activities for each context, family, and child needs. The following statement exemplifies the presentation of the activity by the home visitor to the caregiver and the child and the incentive for them to carry out the activity at other times:

> *"The home visitor plays with (CH), explains the activity to (CH) and to me, and how we should continue the activity. Sometimes, (CH) doesn't want to do the activity then, but we try because it's good for him and his developmental [. . .] We usually don't think of activities like those developed by the home visitor." (DF_24)*

While a few caregivers understood that child development happens in stages, most did not have much knowledge on the stages of ECD. About eight caregivers perceived, through participation in the PCF, the importance of playing with their children to promote ECD. In fact, caregivers were surprised by the activities their children were able to accomplish when proposed by the home visitors. From the activities proposed by the home visitors, caregivers began to observe the children, to understand their preferences, and to learn how to stimulate them through play and by taking the time with them to watch television or read a book, for example. Similarly, pregnant women reported understanding the importance of interacting with the baby while still in the womb. Therefore, all caregivers shared the same vision that the activities of the PCF could support their child's development. The following statement exemplifies caregivers' understanding of the importance of early stimulation and play to promote ECD:

> *"We always repeat the activities. My husband says, 'If we only do it once per week with the home visitor, (CH1) won't develop.' So, we do a lot of things with our son."* (DF_48)

## Safety and security

The subthemes reported by the families were the socioeconomic vulnerability of the families, the environments in which they live, the multisectoral actions carried out by the PCF, the social support of the community, and the dissemination of the PCF.

At least fifteen caregivers reported living in homes with safe and protected spaces for their children to play and grow as well as the existence of outside spaces within their communities such as playgrounds and skate parks. However, almost 2/3 caregivers were unemployed and had no income to meet the food and education needs of their children. They shared concerns about housing instability such as lack of enough money to pay the rent and water, electricity, and gas bills. Some families reported that the cash transfer program alleviated part of these issues. However, not all families were eligible or have access to these social welfare programs. In addition, caregivers reported experiencing stressful situations such as an unwanted pregnancy, the death of a father or close relatives, accidents and house floods, and the prison of the mother or father that changed the dynamics of the family, increasing their socioeconomic and emotional vulnerability. The following statement illustrates the stressful situations experienced by a family and demonstrates their emotional vulnerability:

*"Several factors happened in my life that impacted our finances. The discovery of my daughter's illness, an unwanted pregnancy, [. . .] the death of my grandfather. [. . .] We still don't [have] electricity because it was disconnected due to missing payment. I have accepted this pregnancy because I didn't want to blame her. I know it's not the child's fault, but it was a lot for me to take in: separation from my husband, a new pregnancy, a child, financial difficulties, and psychological problems" (DF_44).*

Caregivers appreciated when PCF home visitors provided information about the social assistance services and programs that families may be eligible for. Through the education and referrals provided by home visitors, some families began to receive the conditional cash transfer benefit and other social welfare benefits. However, the approval to get these benefits was reported to be challenging and slow. In addition, often these benefits are temporary, such as in the case of the Full Plate program (*Prato Cheio)*, which lasted for only six months. The following statement exemplifies the actions of home visitors as facilitators for accessing social welfare benefits:

*"I have regular meetings (every three months) at the CRAS [. . .] and it's very good [. . .]the first one I went to [. . .] was a meeting with the CRAS manager, where he explained all the social welfare benefits, including how each worked, which benefits were responsibility of the CRAS and which benefits were from the Ministry of Education [. . .]the second meeting was with a psychologist and was more a dynamic targeting at mothers participating in the PCF."* (DF_43)

In emergency situations such as lack of food or other survival supplies, caregivers felt they could not rely on social welfare benefits. Thus, families reported seeking social support from non-governmental sources such as religious institutions, community, or friends/family that assist them with food baskets and boxes with baby supplies.

## Discussion

Our study documented that PCF enhanced the five components of the nurturing care environment provided by families enrolled in the program for at least six months. Furthermore, we identified complex vulnerabilities and external contextual factors, such as food insecurity and socioeconomic needs, which require PCF multisectoral actions to achieve the program's ultimate goal of promoting optimal childhood development. Caregivers with less access to financial resources are a predictor of lower childhood development outcomes, especially child cognitive development, growth, and nutrition [21]. Evidence shows that socioeconomic vulnerabilities, such as food insecurity [22, 23] and maternal mental health, are linked with disparities in ECD [24–27]. Corroborating with a prior study [21], we found that poor maternal mental health (i.e., depression and anxiety) as well as the burnout caused by excess responsibilities of childcare and household chores are exacerbated by the lack of support and/or the presence of a partner. A study conducted in Brazil showed that low-income mothers without a partner, compared to mothers with a partner, are more likely to have anxiety and a lower perception of their quality of life, which can be aggravated when these women live in extreme poverty [28]. In this context, playing or replicating the early stimulation activities with the child may increase parental stress and the burden for the main caregiver. However, our study indicated that despite the burnout caregivers were sensitized to the importance of the early stimulation activities to promote ECD and reported responsive practices that were previously not common in the family's routine. Similarly, caregivers participating in the PCF in six Brazilian municipalities demonstrated increased knowledge about ECD and responsive caregiving. [29].

Our study found a low participation of fathers during PCF activities, which has been found in another study evaluating PCF home visits [29]. Father's involvement is associated with overall improved ECD (cognitive, language, and social-emotional skills, as well as physical and mental health) [30, 31]. Evidence shows that engagement of fathers during a child development decreases depression, fear, criminality, and substance abuse into adulthood [31, 32]. Additionally, father involvement is beneficial to women's health [32]. In this context, incentivizing the participation of fathers during home visits could increase the impact of PCF activities.

Caregivers' perceptions of the home visitors were positive regarding the flexibility, appropriateness, and consistency of visits, contrary to what was reported in the qualitative study conducted by the Ministry of Citizenship in 2020 [29]. Our findings highlight the importance of the bond created between caregivers and home visitors and between children and home visitors. In this context, the use of the Care for Child Development methodology [33, 34] helped home visitors to identify the child's interests and to perform the activities respecting their will as well as to help the families to use materials available at home. This methodology encourages responsive caregiving practices, improves the quality of interactions between caregivers and children, helps caregivers provide nurturing care, and improves child developmental outcomes [34]. On the other hand, our study identified that home visitors must be prepared to address and provide support on issues adjacent to ECD, including maternal mental health, exclusive breastfeeding, disciplinary practices, father engagement, as well as educate and refer families to social welfare programs and other programs accordingly to their needs. Therefore, it is critical to provide comprehensive training for PCF home visitors and supervisors that cuts across all components of the nurturing care environment to address the needs of the families [10, 29].

Operationalizing PCF nurturing care multisectoral actions has been a major challenge during the implementation of the program [10]. Nurturing care multisectoral actions within the scope of the PCF are an important strategy to facilitate and meet families' needs and to increase the impact of home visits on and early stimulation of ECD. In our study, multisectoral referrals done by home visitors were mostly related to social welfare benefits. Evidence has proven that combining conditional social welfare benefits such as cash transfer programs with parenting programs improved responsive parenting practices and early childhood cognition and language skills [35]. However, our study identified the limited scope of multisectoral actions, which may influence the low impact of PCF on long-term ECD outcomes [11]. Therefore, the integration of PCF with multiple sectors within each catchment area through the Multisectoral Early Childhood Centers may be a powerful strategy to enhance the operationalization of multisectoral nurturing care actions.

Our study was designed to identify the influence of PCF on the nurturing care environment from the perceptions of caregivers in the DF. The chosen sample is a strong point of this study, as it included all groups eligible (pregnant women, children up to three years old, whose families are beneficiaries of the Bolsa Família Program and children up to six years old whose families are beneficiaries of the Continuous Cash Benefit–BPC) for the PCF that expressed the perspectives necessary to achieve the aim of this study, even though the study did not include representatives from all the administrative regions of DF where the PCF has been implemented. The Nurturing Care Framework is a consolidated guide addressed to stakeholders and governments with five components that can support the development of children.

In summary, the PCF enhanced the components of the nurturing care environment provided by families. The identified benefits include improving the child's mental health and promoting responsive parenting practices. The bond between the family and the home visitor is critical for the success of the intervention. However, families' vulnerabilities and external contextual implementation barriers may prevent families from fully benefiting from PCF

activities. Therefore, enhancing the multisectoral component of the PCF to address topics relevant to families could further help enable a nurturing care environment.

## Supporting information

**S1 Appendix. Description of the Programa Criança Feliz (PCF) in Brasilia, Brazil: Principles and governance.** Customization and modification of TIDieR items were not applicable to this description.
(DOCX)

**S2 Appendix. In-depth interview guide.** Translated from the original Portuguese version.
(DOCX)

**S3 Appendix. Themes and subthemes identified in interviews with families participating in the Programa Criança Feliz in the Federal District, Brazil.**
(DOCX)

## Acknowledgments

This article analyzes data from the project evaluating the implementation of the Brazil's Criança Feliz Program in the Federal District led by the Department of Social and Behavioral Health at the University of Nevada, Las Vegas in collaboration with the Postgraduate Program in Human Nutrition of the University of Brasília. The quotes from the interviews were translated by Alene Alder-Rangel of Alder's English Services.

## Author Contributions

**Conceptualization:** Laura Mendes Toledo Dal'Ava dos Santos, Gabriela Buccini.

**Data curation:** Lidia Godoi, Gabriela Buccini.

**Formal analysis:** Laura Mendes Toledo Dal'Ava dos Santos, Beatriz de Andrade e Guimarães, Isabela Mendes Coutinho, Gabriela Buccini.

**Funding acquisition:** Vivian S. S. Gonçalves, Gabriela Buccini.

**Investigation:** Lidia Godoi.

**Project administration:** Nathalia Pizato, Vivian S. S. Gonçalves.

**Supervision:** Nathalia Pizato, Vivian S. S. Gonçalves, Gabriela Buccini.

**Writing – original draft:** Laura Mendes Toledo Dal'Ava dos Santos, Gabriela Buccini.

**Writing – review & editing:** Lidia Godoi, Beatriz de Andrade e Guimarães, Isabela Mendes Coutinho, Nathalia Pizato, Vivian S. S. Gonçalves, Gabriela Buccini.

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
