## [Decision Letter · Decision Letter 0]

31 May 2023

PONE-D-23-10075A qualitative analysis of the nurturing care environment of families participating in Brazil’s Criança Feliz early childhood programPLOS ONE

Dear Dr. Buccini,

Thank you for submitting your manuscript to PLOS ONE. After careful consideration, we feel that it has merit but does not fully meet PLOS ONE’s publication criteria as it currently stands. Therefore, we invite you to submit a revised version of the manuscript that addresses the points raised during the review process. The authors presented an interesting paper. I include some comments bellow. It will be important to consider the comments made by the reviewers.

We look forward to receiving your revised manuscript.

Kind regards,

Angelica Espinosa Miranda, M.D., Ph.D.

Academic Editor

PLOS ONE

Journal Requirements:

2. a) Please include a complete copy of PLOS’ questionnaire on inclusivity in global research in your revised manuscript. Our policy for research in this area aims to improve transparency in the reporting of research performed outside of researchers’ own country or community. The policy applies to researchers who have travelled to a different country to conduct research, research with Indigenous populations or their lands, and research on cultural artefacts. The questionnaire can also be requested at the journal’s discretion for any other submissions, even if these conditions are not met.  Please find more information on the policy and a link to download a blank copy of the questionnaire here: https://journals.plos.org/plosone/s/best-practices-in-research-reporting. Please upload a completed version of your questionnaire as Supporting Information when you resubmit your manuscript.

b) In the ethics statement in the Methods, you have specified that verbal consent was obtained. Please provide additional details regarding how this consent was documented and witnessed, and state whether this was approved by the IRB

   "L. M. T. D. Santos received scholarship from the National Council for Scientific and Technological Development (CNPQ). VSSG received support for the data analysis of the interviews carried out in the Federal District received partial funding from the Federal District Research Support Foundation (FAPDF) under number 498/2021 - FAPDF/SUCTI/COOTEC. GB received support from the Eunice Kennedy Shriver National Institute of Child Health & Human Development of the National Institutes of Health under Award Number R00HD097301. The research reported in this publication was supported by The content is the sole responsibility of the authors and does not necessarily represent the official opinion of the National Institutes of Health. "

Additional Editor Comments:

It is an interesting article about an important topic. The authors could offer more information about the process of approach and inclusion of participants. How were they approached? When was the project explained and asked to sign the consent form? Were there refusals to participate? Did the authors use a qualitative Data Analysis Software for performing the analysis? It will be important to verify the format of the references according to the rules recommended by the journal.

Reviewers' comments:

Reviewer's Responses to Questions

**Comments to the Author**

1. Is the manuscript technically sound, and do the data support the conclusions?

Reviewer #1: Yes

Reviewer #2: Yes

2. Has the statistical analysis been performed appropriately and rigorously? 

Reviewer #1: Yes

Reviewer #2: N/A

3. Have the authors made all data underlying the findings in their manuscript fully available?

Reviewer #1: Yes

Reviewer #2: No

4. Is the manuscript presented in an intelligible fashion and written in standard English?

Reviewer #1: Yes

Reviewer #2: Yes

5. Review Comments to the Author

Reviewer #1: The Nurturing Care Framework is a considated guide addressed to can suport the development of children. This manuscript does an good job demostrating how goverment support low income families in Brazil.

There is a revision to provider.

I suggest to the author describing the "ethical issues" at the end of the chapter "Study design an ethical aspects".

I recommend to the author review "References", for exemplo, 26 and 28 numbers and perform adequacy according to the norms.

Reviewer #2: A proposta de resumo está bem posta e apresenta dados relevantes para o entendimento do que se propõe o artigo.

A contextualização, a importância social do projeto PCF, o gap identificado e o objetivo do artigo estão bem postos na introdução.

Uma sugestão seria que a partir da linha 67 até 74, pudesse ser já antecipado ao leitor uma breve justificativa sobre o motivo de o estudo ficar restrito ao Distrito Federal (DF), já que alcançou 3.028, municípios do Brasil. Por ser um estudo qualitativo, vamos entender isso na metodologia, mas seria importante já se antecipar esse argumento brevemente na introdução.

Sobre o método, o recorte da pesquisa realizada no Distrito Federal, o modo como foram selecionados os participantes da pesquisa e as questões éticas levadas em conta, acredito que estão bem postas. Também o detalhamento do da elaboração do guia de entrevistas e a forma como foram coletados e gestionados os dados estão bem apresentados no artigo.

A forma como foram sistematizados os dados para análise também está explicitada no artigo, contudo, seria interessante se tívessemos um gráfico ou ilustração que explicitasse o processo de saturação de dados mencionada a partir da linha 144 e seguintes. Do ponto de vista metodológico, compartilhar o livro de códigos produzido durante a pesquisa e que foi aperfeiçoado durante a investigação (se poderia explicitar como ocorreu esse aperfeiçoamento?) . Creio que são dados metodológicos muito ricos que poderiam deixar mais caracterizado o processo qualitativo da investigação.

Por fim, em relação aos resultados, vemos como qualitativamente significativos. Possuem uma importância enorme para entendermos a situação em que se encontra os sujeitos da pesquisa em toda fase do processo do Programa e que nos fazem perceber seus méritos e problemas a serem melhorados. Contudo, embora sejam 22 entrevistados, talvez se pudesse informar quantos dos entrevistados em cada ponto, de fato, podem ser citados como relacionados às categorias destacadas e exemplificadas com trechos das entrevistas.

Por exemplo, linha 182 (Em geral, os prestadores...); linha 188 (A maioria dos prestadores...); linha 202 (De fato, muitos prestadores de cuidado referiram..."; entre outros momentos. Talvez houvesse uma forma de dimensionar melhor esse quantitativo, que embora não sirva como estatística, serviria para dimensionar melhor o universo das percepções e situações apresentadas. Aqui, temos para um exercício futuro, um conjunto de aspectos que poderiam dimensionar uma pesquisa nacional quantitativa que poderia dar robustez, fortaleza e capacidade de generalização para estas importantes categorias identificadas na pesquisa.

Por fim, a discussão é pertinente, bem articulada e a bibliografia é atualizada, apresentando considerações que são relevantes para o desenvolvimento de melhorias das políticas públicas correlacionadas com esta iniciativa do PCF.

6. PLOS authors have the option to publish the peer review history of their article (what does this mean?). If published, this will include your full peer review and any attached files.

Reviewer #1: No

Reviewer #2: No

---

## [Author Response · Author response to Decision Letter 0]

19 Jun 2023

Dear Dr. Angelica Espinosa Miranda

Academic Editor, Plos One

We thank the reviewers for their review and valuable feedback. We replied carefully to each comment which resulted in an improved version for your consideration. We hope the changes meet your expectations.

Thank you for the opportunity. 

Authors.

Additional Editor Comments:

It is an interesting article about an important topic. 

Response: Thank you for your feedback.

The authors could offer more information about the process of approach and inclusion of participants. How were they approached? When was the project explained and asked to sign the consent form? 

Response: Thank you for this question. Participants interviewed complied the predetermined inclusion criteria in this research and were randomly selected from a list of families enrolled in the program at the time of the interview (described page 6, line 121).

Were there refusals to participate? 

Response: There was no refusals to participate, as stated on page 6, line 129.

Did the authors use a qualitative Data Analysis Software for performing the analysis? It will be important to verify the format of the references according to the rules recommended by the journal.

Response: The authors did not use a qualitative data analysis software for performing the analysis. 

Review Comments to the Author

Reviewer #1: 

The Nurturing Care Framework is a considated guide addressed to can suport the development of children. This manuscript does an good job demostrating how goverment support low income families in Brazil.

Response: Thank you for your feedback.

There is a revision to provider. I suggest to the author describing the "ethical issues" at the end of the chapter "Study design an ethical aspects".

Response: As suggested, the following information has been added ‘Following a description of the aim and design of the study, all participants provided verbal informed consent, which was audio recorded at the beginning of the in-depth interview. The verbal consent was approved by both the Brazilian Ethics Committee and UNLV IRB. All participants received a copy of the consent form by virtual means.’ (p.5, l. 91-94)

I recommend to the author review "References", for exemplo, 26 and 28 numbers and perform adequacy according to the norms.

Response: As suggested, the reference list was revised. Please see below a list of revisions made: 

p.22, l.559 : :1-55.

p.22, l.563 : 1 :34-48

p.22, l.566 : :1506

p.22, l.574 : Diário Oficial da União 5 oct 2016.

p.22, l. 577-578 : Brasil [homepage of internet]. O Criança Feliz. Brasil: Ministério da Cidadania [acess 10 jan 2023].

p.23, l. 645: Women’s Health. 2019 ;15:1 –55.

p.24, l.653 : 2020;1-110 [acess 05 jan 2023]. Available from: https://www.plan-eval.com/repositorio/arquivo/artigo/Programa_Criança_Feliz_150121.pdf

Reviewer #2: 

A proposta de resumo está bem posta e apresenta dados relevantes para o entendimento do que se propõe o artigo.A contextualização, a importância social do projeto PCF, o gap identificado e o objetivo do artigo estão bem postos na introdução.

Response: Thank you for your feedback.

Uma sugestão seria que a partir da linha 67 até 74, pudesse ser já antecipado ao leitor uma breve justificativa sobre o motivo de o estudo ficar restrito ao Distrito Federal (DF), já que alcançou 3.028, municípios do Brasil. Por ser um estudo qualitativo, vamos entender isso na metodologia, mas seria importante já se antecipar esse argumento brevemente na introdução.

Response: As suggested, we included information about the Distrito Federal in the background. 

p.4 l.69-73

Sobre o método, o recorte da pesquisa realizada no Distrito Federal, o modo como foram selecionados os participantes da pesquisa e as questões éticas levadas em conta, acredito que estão bem postas. Também o detalhamento do da elaboração do guia de entrevistas e a forma como foram coletados e gestionados os dados estão bem apresentados no artigo.

Response: Thank you for your feedback.

A forma como foram sistematizados os dados para análise também está explicitada no artigo, contudo, seria interessante se tívessemos um gráfico ou ilustração que explicitasse o processo de saturação de dados mencionada a partir da linha 144 e seguintes. Do ponto de vista metodológico, compartilhar o livro de códigos produzido durante a pesquisa e que foi aperfeiçoado durante a investigação (se poderia explicitar como ocorreu esse aperfeiçoamento?). Creio que são dados metodológicos muito ricos que poderiam deixar mais caracterizado o processo qualitativo da investigação.

Response: Thank you for your feedback. We created and added a figure to summarize the data analysis steps taken to iteratively refine the codebook. The following statement was added to the data analysis section: Fig 1. outlines the data analysis steps to iteratively refine the codebook.

Fig 1. Illustration of data analysis steps to create and refine the codebook.

Por fim, em relação aos resultados, vemos como qualitativamente significativos. Possuem uma importância enorme para entendermos a situação em que se encontra os sujeitos da pesquisa em toda fase do processo do Programa e que nos fazem perceber seus méritos e problemas a serem melhorados. Contudo, embora sejam 22 entrevistados, talvez se pudesse informar quantos dos entrevistados em cada ponto, de fato, podem ser citados como relacionados às categorias destacadas e exemplificadas com trechos das entrevistas.

Por exemplo, linha 182 (Em geral, os prestadores...); linha 188 (A maioria dos prestadores...); linha 202 (De fato, muitos prestadores de cuidado referiram..."; entre outros momentos. Talvez houvesse uma forma de dimensionar melhor esse quantitativo, que embora não sirva como estatística, serviria para dimensionar melhor o universo das percepções e situações apresentadas. Aqui, temos para um exercício futuro, um conjunto de aspectos que poderiam dimensionar uma pesquisa nacional quantitativa que poderia dar robustez, fortaleza e capacidade de generalização para estas importantes categorias identificadas na pesquisa.

Response: As suggested, we included the following excerpts: 

p.9, l.194 : ‘All the caregivers reported accessing primary healthcare services prior to enrolling in the PCF.”

p.10, l.209: ‘More than half of the interviewees reported exhaustion,[…]’

p.10, l.211: ‘At least two caregivers reported[…]’

p.11, l.242: ‘Half of the caregivers[…]”

p.12, l.271: ‘According to eight caregivers[…]”

p.12, l.273: ‘At least two families reported that the fathers[…]”

p.13, l.279: ‘About five caregivers wished to[…]”

p.14, l.308 : ‘Four caregivers reported using screens[…]’

p.15, l.328: ‘However, three families reported barriers that prevent the children[…]’

p.15, l.341: ‘About eight caregivers perceived, […]’

p.16, l.358: ‘At least fifteen caregivers reported living in homes with safe and protected spaces […]’

p.16, l.360: ‘However, almost 2/3 of caregivers […]’

Por fim, a discussão é pertinente, bem articulada e a bibliografia é atualizada, apresentando considerações que são relevantes para o desenvolvimento de melhorias das políticas públicas correlacionadas com esta iniciativa do PCF.

Response: Thank you for your feedback.

---

## [Editor Report · Decision Letter 1]

7 Jul 2023

A qualitative analysis of the nurturing care environment of families participating in Brazil’s Criança Feliz early childhood program

PONE-D-23-10075R1

Dear Dr.Gabriela Buccini ,

We’re pleased to inform you that your manuscript has been judged scientifically suitable for publication and will be formally accepted for publication once it meets all outstanding technical requirements.

Kind regards,

Angelica Espinosa Miranda, M.D., Ph.D.

Academic Editor

PLOS ONE
---

## [Editor Report · Acceptance letter]

11 Jul 2023

PONE-D-23-10075R1 

A qualitative analysis of the nurturing care environment of families participating in Brazil’s *Criança Feliz* early childhood program 

Dear Dr. Buccini:

I'm pleased to inform you that your manuscript has been deemed suitable for publication in PLOS ONE. Congratulations! Your manuscript is now with our production department. 

Kind regards, 

on behalf of

Dr. Angelica Espinosa Miranda 

Academic Editor

PLOS ONE